# A Mixture of Surprises for Unsupervised Reinforcement Learning

**Andrew Zhao** [1]*     **Matthieu Gaetan Lin** [2]*     **Yangguang Li**[3]     **Yong-Jin Liu**[2†]

**Gao Huang** [1†]

[1] Department of Automation, BNRist, Tsinghua University
[2] Department of Computer Science, BNRist, Tsinghua University
[3] SenseTime

{zqc21,lyh21}@mails.tsinghua.edu.cn,
liyangguang@sensetime.com
{liuyongjin,gaohuang}@tsinghua.edu.cn,

## Abstract

Unsupervised reinforcement learning aims at learning a generalist policy in a reward-free manner for fast adaptation to downstream tasks. Most of the existing methods propose to provide an intrinsic reward based on surprise. Maximizing or minimizing surprise drives the agent to either explore or gain control over its environment. However, both strategies rely on a strong assumption: the entropy of the environment's dynamics is either high or low. This assumption may not always hold in real-world scenarios, where the entropy of the environment's dynamics may be unknown. Hence, choosing between the two objectives is a dilemma. We propose a novel yet simple mixture of policies to address this concern, allowing us to optimize an objective that simultaneously maximizes and minimizes the surprise. Concretely, we train one mixture component whose objective is to maximize the surprise and another whose objective is to minimize the surprise. Hence, our method does not make assumptions about the entropy of the environment's dynamics. We call our method a **M**ixture **O**f **S**urprise**S** (MOSS) for unsupervised reinforcement learning. Experimental results show that our simple method achieves state-of-the-art performance on the URLB benchmark, outperforming previous pure surprise maximization-based objectives. Our code is available at: `https://github.com/LeapLabTHU/MOSS`.

## 1 Introduction

Humans can learn meaningful behaviors without external supervision, *i.e.*, in an unsupervised manner, and then adapt those behaviors to new tasks [3]. Inspired by this, unsupervised reinforcement learning decomposes the reinforcement learning (RL) problem into a pretraining phase and a finetune phase [29]. During a pretraining phase, an agent prepares all possible tasks that a user might select. Afterward, the agent tries to figure out the selected task as quickly as possible during finetuning [22]. Doing so allows solving the RL problem in a meaningful order [3], *e.g.*, a cook first has to look at what is in the fridge before deciding what to cook. Unsupervised representation learning has shown great success in computer vision [25] and natural language processing [9]; however, one challenge is that RL includes both behavior learning and representation learning [29, 49].

---

*Equal contribution.
†Corresponding authors.

36th Conference on Neural Information Processing Systems (NeurIPS 2022).

Current unsupervised RL methods provide an intrinsic reward to the agent during a pretraining phase to tackle the behavior learning problem [29]. Intuitively, this intrinsic reward should incentivize the agent to understand its environment [13]. Current methods formulate the intrinsic reward as either maximizing or minimizing surprise [41, 6, 33, 52, 38, 39, 10, 32, 28, 15, 43, 42, 12], where knowledge-based methods quantify surprise as the uncertainty of a prediction model [10, 38, 39] and data-based methods measure surprise as an information-theoretic quantity [41, 6, 52, 33, 5][3]. Surprise maximization methods [22, 28, 33] formulate the problem as an exploration problem. Intuitively, to prepare all possible downstream tasks, an agent has to explore the state space and figure out what is possible in the environment. For instance, one instantiation is to maximize the agent's state entropy [33]. In contrast to surprise maximization, another line of work takes inspiration from the free-energy principle [20, 19, 18] and proposes to minimize the surprise [6, 41]. In particular, these works argue that external perturbations naturally provide the agent with surprising events. Hence, an agent can learn meaningful behaviors by minimizing its surprise, and minimizing this quantity requires gaining control over these external perturbations [41, 6]. For example, SMiRL [6] proposed to minimize the agent's state entropy.

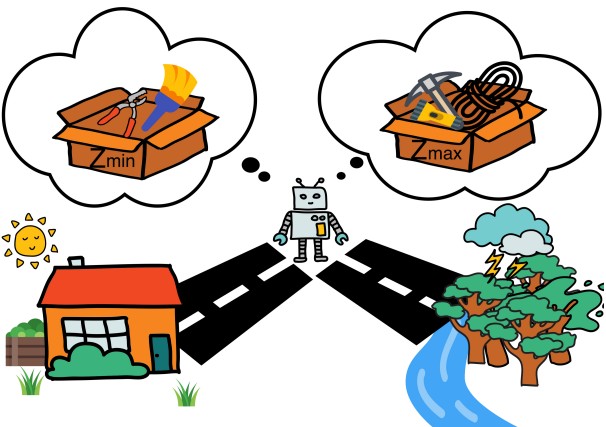

Figure 1: **Outline of our Mixture Of SurpriseS (MOSS) strategy**. We provide the agent with two paths, each corresponding to maximizing or minimizing surprise. During the pretraining phase, the agent gathers experience from both paths using the skills from the corresponding toolbox. Finally, during the finetuning phase, the agent can choose any skills from any toolbox.

A closer look at these two approaches indicates a strong assumption. Substantial external perturbations already naturally provide an agent with a high state entropy [6]. Therefore, surprise-seeking approaches assume that the environment does not provide significant external perturbations. On the other hand, in an environment without any external perturbations, the agent already achieves minimum entropy by not acting [41]. Therefore, surprise minimization approaches assume that the environment provides external perturbations for the agent to control. However, in real-world scenarios, it is often difficult to quantify the entropy of the environment's dynamics beforehand, or the agent might face both settings. For example, a butler robot performs mundane chores daily in a household. However, the robot might also encounter surprising events (*e.g.*, putting out a fire). Therefore a fixed assumption on the entropy of the environment dynamics is often not possible. In other words, choosing between surprise maximization or minimization for pretraining is a dilemma.

Simultaneously optimizing these two opposite objectives does not make sense. Instead, in this paper, we show that competence-based methods [29] offer a simple yet effective way to combine the benefits of these two objectives. In addition to conditioning the policy on the state, competence-based methods condition the policy on a latent vector [28, 32, 24, 50, 21, 1, 16]. Conditioning the policy offers an appealing way to formulate the unsupervised RL problem. During a pretraining phase, the agent tries to learn a set of possible behaviors in the environment and distills them into skills. Then, during finetuning, the agent tries to figure out the selected task as quickly as possible, using the repertoire of skills gathered during pretraining. For example, as illustrated in Fig. 1, we view skill distributions as a mixture of policies instead of a single policy. In particular, we train one set of skills whose objective is to maximize the surprise and another set of skills whose objective is to

---

[3]Refer to [29] for detailed definitions of data-based, knowledge-based and competence-based methods.

minimize the surprise. Surprisingly, this paper shows that this simple approach works well in practice, which simultaneously optimizes two contradicting objectives. Our primary contribution, presented in Section 4, is a simple intrinsic reward called MOSS that does not make assumptions about the entropy of the environment's dynamics. In Section 5, our experimental results on URLB [29] and ViZDoom [27] show that, surprisingly, our MOSS methods achieve state-of-the-art results.

We organize the paper as follows. Section 3 briefly analyzes previous unsupervised RL algorithms under the surprise framework. Then in Section 4, we introduce our MOSS method. Next, experimental results in Section 5 shows that on URLB [29] and ViZDoom [27], our MOSS method improves upon previous pure maximization and minimization methods. Finally, we provide discussions and limitations in Section 6.

## 2 Preliminaries

**Markov Decision Process.** Unsupervised RL methods studied in this paper operate under a Markov Decision Process (MDP) [47]. In particular, we specify an MDP as a tuple $\mathcal{M} = (\mathcal{S}, \mathcal{A}, T, r^{\text{ext}}, \rho, \gamma)$ where $\mathcal{S}$ is the state space and $\mathcal{A}$ is the action space of the environment. $T(\mathbf{S}' \mid \mathbf{S}, \mathbf{A})$ represents the state-transition dynamics, $\rho : \mathcal{S} \to [0, 1]$ is the initial state distribution, and $\gamma \in [0, 1)$ is the discount factor. At each discrete time step $t \in \mathbb{Z}^*$, the agent receives a state and performs an action, which we denote as $\mathbf{S}_t \in \mathcal{S}$ and $\mathbf{A}_t \in \mathcal{A}$, respectively. During pretraining, unsupervised RL algorithms compute an intrinsic reward $r^{\text{int}}$; during the finetune phase, the agent receives the extrinsic reward $r^{\text{ext}}$ given by the environment at each interaction.

**Skill.** Intuitively, a *skill* is an abstraction of a specific behavior (*e.g.*, walking), and in practice, a *skill* is a latent conditioned policy [15]. Given a latent vector $\mathbf{z}$, we denote a skill as $\pi_\theta(\mathbf{a}_t \mid \mathbf{s}_t, \mathbf{z})$, where $\pi_\theta$ is the policy parameterized by $\theta$. For instance, during pretraining, the latent vectors are sampled every $n$ steps such that the latent vector $\mathbf{z}$ is associated with the behavior executed during the associated $n$ steps.

**Mutual Information.** Knowledge-based, data-based, and competence-based methods have different measures of surprise. The study in this paper falls into the category of competence-based methods. In particular, data-based and competence-based methods rely on an information-theoretic definition of surprise, *i.e.*, entropy. Previous competence-based methods acquire skills by maximizing mutual information [14] between $\mathcal{T}$ and skills $\mathbf{Z}$

$$\mathbb{I}(\mathcal{T}; \mathbf{Z}) = \mathbb{H}[\mathcal{T}] - \mathbb{H}[\mathcal{T}|\mathbf{Z}] \tag{1}$$
$$= \mathbb{H}[\mathbf{Z}] - \mathbb{H}[\mathbf{Z}|\mathcal{T}], \tag{2}$$

where $\mathcal{T}$ can be the states $\mathbf{S}$, the joint distribution of state-transitions $(\mathbf{S}', \mathbf{S})$, or the state-transitions $(\mathbf{S}' \mid \mathbf{S})$. In particular, these methods differ in how they decompose the mutual information. Theoretically, these different decompositions are equivalent, *i.e.*, they all maximize the mutual information between states and skills. However, the particular choice greatly influences the performance in practice as optimizing this objective relies on approximations.

To motivate the potential of competence-based methods over data-based or knowledge-based methods, we provide an intuitive understanding of Eq. (1). On the one hand, the entropy term says that we want skills in aggregate that explore the state space; we use it as a proxy to learn skills that cover the set of possible behaviors. On the other hand, It is not enough to learn skills that randomly go to different places. We want to reuse those skills as accurately as possible, meaning we need to be able to discriminate or predict the agent's state transitions from skills. To do so, we minimize conditional entropy. In other words, appropriate skills should cover the set of possible behaviors and should be easily distinguishable.

## 3 Information-Theoretic Skill Discovery

Competence-based methods employ different intrinsic rewards to maximize mutual information: (1) discriminability-based and (2) exploratory-based intrinsic rewards. For example, the former rewards the agent for discriminable skills. In contrast, the latter rewards the agent for skills that effectively cover the state space using a KNN density estimator [44, 33] to approximate the entropy term. Below we analyze both approaches.

## 3.1 Discriminability-based Intrinsic Reward

Previous work such as DIAYN [15] uses a discriminability-based intrinsic reward. They use the decomposition in Eq. (2) and given a variational distribution $q_\phi$, they optimize the following variational lower bound

$$
\begin{aligned}
\mathbb{I}(\mathbf{S}; \mathbf{Z}) &= \mathbb{KL}(p(\mathbf{s}, \mathbf{z}) \ || \ p(\mathbf{s})p(\mathbf{z})) \\
&= \mathbb{E}_{\mathbf{s}, \mathbf{z} \sim p(\mathbf{s}, \mathbf{z})} \left[ \log \frac{q_\phi(\mathbf{z} \mid \mathbf{s})}{p(\mathbf{z})} \right] + \mathbb{E}_{\mathbf{s} \sim p(\mathbf{s})}[\mathbb{KL}(p(\mathbf{z} \mid \mathbf{s}) \ || \ q_\phi(\mathbf{z} \mid \mathbf{s}))] \\
&\geq \mathbb{E}_{\mathbf{z}, \mathbf{s} \sim p(\mathbf{z}, \mathbf{s})}[\log q_\phi(\mathbf{z} \mid \mathbf{s})] - \mathbb{E}_{\mathbf{z} \sim p(\mathbf{z})}[p(\mathbf{z})],
\end{aligned}
$$

where $\mathbf{Z} \sim p(\mathbf{Z})$ is a discrete random variable. In particular, the discriminator rewards the agent if it can guess the skill from the state, *i.e.*, $r^{\text{int}}(\mathbf{s}) \triangleq \log q_\phi(\mathbf{z} \mid \mathbf{s}) - \log p(\mathbf{z})$. These methods may easily run into a chicken and egg problem, where skills learn to be diverse using the discriminator's output. However, the discriminator cannot learn to discriminate skills if the skills are not diverse. Hence, it discourages the agent from exploring. Previous work [45] has tried to solve this problem by decoupling the *aleatoric uncertainty* from the *epistemic uncertainty*.

Furthermore, solving the chicken and egg problem is not enough since a state must map to a single skill in Eq. (2). Accordingly, the methods that use the decomposition in Eq. (2) require the skill space to be smaller than the state space so that the skills are distinguishable. Since previous work showed that a high-dimensional continuous skill space empirically performs better, the intrinsic reward should not rely on the decomposition in Eq. (2). Intuitively, a continuous skill space allows (1) to interpolate between skills and (2) to represent a more significant set of skills in a more compact representation.

Instead, in Eq. (1), in contrast to skills that are predictable by states $\mathbb{H}[\mathbf{Z} \mid \mathcal{T}]$, we require that states are predictable by skills $\mathbb{H}[\mathcal{T} \mid \mathbf{Z}]$. Since any state should be predictable from a given skill, the skill space must be larger than the state space (*i.e.*, we do not want a skill mapping to multiple states). Therefore, another work [43] relies on a variational bound on Eq. (1).

$$
\begin{aligned}
\mathbb{I}(\mathbf{S}'; \mathbf{Z} \mid \mathbf{S}) &= \mathbb{E}_{\mathbf{z}, \mathbf{s}, \mathbf{s}' \sim p(\mathbf{z}, \mathbf{s}, \mathbf{s}')} \left[ \log \frac{q_\phi(\mathbf{s}' \mid \mathbf{s}, \mathbf{z})}{p(\mathbf{s}' \mid \mathbf{s})} \right] + \mathbb{E}_{\mathbf{z}, \mathbf{s} \sim p(\mathbf{z}, \mathbf{s})} \left[ \mathbb{KL}(p(\mathbf{s}' \mid \mathbf{s}, \mathbf{z}) \ || \ q_\phi(\mathbf{s}' \mid \mathbf{s}, \mathbf{z})) \right] \\
&\geq \mathbb{E}_{\mathbf{z}, \mathbf{s}, \mathbf{s}' \sim p(\mathbf{z}, \mathbf{s}, \mathbf{s}')} \left[ \log \frac{q_\phi(\mathbf{s}' \mid \mathbf{s}, \mathbf{z})}{p(\mathbf{s}' \mid \mathbf{s})} \right],
\end{aligned}
$$

which uses a continuous skill space; however, it does not scale to high dimensions as the intrinsic reward $r^{\text{int}}(\mathbf{s}', \mathbf{a}, \mathbf{s}) \triangleq \log \frac{q_\phi(\mathbf{s}' \mid \mathbf{s}, \mathbf{z})}{p(\mathbf{s}' \mid \mathbf{s})}$ relies on an estimation of $p(\mathbf{s}' \mid \mathbf{s})$. In particular, they assume that $p(\mathbf{z} \mid \mathbf{s}) = p(\mathbf{z})$. Intuitively, given $\mathbf{s}$, if we assume each element in the latent vector $(\mathbf{z})_i$ to be independent of each other, the error of this assumption will be scaled by the dimension of $\mathbf{z}$.

## 3.2 Exploratory-based Intrinsic Reward

As aforementioned, discriminability-based intrinsic reward may easily run into a chicken and egg problem. Hence, other work [28, 32] uses an exploratory-based intrinsic reward to address the chicken and egg problem. In other words, they use the decomposition in Eq. (1), where the intrinsic reward explicitly rewards the agent for exploring through $\mathbb{H}[\mathcal{T}]$.

Previous work [33] maximizes the state entropy, *i.e.*, $\mathcal{T} = \mathbf{S}$. However, the authors in [4] argue that it often results in the discriminator simply memorizing the last state of each skill. Instead, CIC [28] proposes to maximize the entropy of the joint distribution of state-transitions (which from now on we refer as joint entropy), *i.e.*, $\mathbb{H}[\mathbf{S}', \mathbf{S}]$.

Therefore, CIC [28] proposes to estimate the conditional entropy $\mathbb{H}[\mathbf{S}', \mathbf{S} \mid \mathbf{Z}]$ using noise contrastive estimation [28, 23] and the joint entropy $\mathbb{H}[\mathbf{S}', \mathbf{S}]$ using a $k$-nearest neighbor estimator [44] to handle high dimensional skill space.

**KNN-density estimation.** Previous works [33, 28] approximate the joint entropy $\mathbb{H}[\mathbf{S}', \mathbf{S}]$ using a $k$-nearest neighbor estimator [44]. Given a random sample of size $N$, $\{(\mathbf{S}^{(i)'}, \mathbf{S}^{(i)})\}_{i=1}^N \sim P(\mathbf{S}', \mathbf{S})$,

the estimator computes a Monte Carlo estimate [7] of the entropy defined as

$$\hat{\mathbb{H}}\big[\mathbf{S}', \mathbf{S}\big] = -\frac{1}{N}\sum_{i=1}^{N}\log \hat{p}_k(\mathbf{s}^{(i)'}, \mathbf{s}^{(i)}) + b(k), \tag{3}$$

where $\hat{p}_k(\mathbf{s}^{(i)'}, \mathbf{s}^{(i)}) = \frac{k}{NV_k^{(i)}}$ is an estimation of $p(\mathbf{s}^{(i)'}, \mathbf{s}^{(i)})$, $V_k^{(i)}$ is the volume of the hypersphere of radius $R_k^{(i)}$, and $b(k)$ is a constant such that the estimator is unbiased. In particular, $R_k$ is the distance from $\mathbf{s}$ to its $k$-nearest neighbor $\mathbf{s}'_k$ where we assume a uniform distribution of points within the hypersphere (local uniformity assumption [34]).

In practice, to make the distance $R_k$ meaningful, we do not operate in the state space. For instance, CIC [28] trains an MLP $f_\psi : \mathbb{R}^{|\mathcal{S}|} \to \mathbb{R}^d$ that projects the state into a latent representation such that $R_k = \|f_\psi(s) - f_\psi(s'_k)\|$ where $d$ is the skill dimension. Moreover, in APT [33], it was found that averaging over all $k$-nearest neighbors leads to better results. Hence, CIC [28] optimizes a quantity proportional to Eq. (3):

$$\hat{\mathbb{H}}\big[\mathbf{S}', \mathbf{S}\big] \triangleq \sum_{i=1}^{N}\log\left(c + \frac{1}{k}\sum_k R_k^{(i)}\right), \tag{4}$$

where $c = 1$ is constant for numerical stability. We view the joint entropy as an expected reward, and for a transition $(\mathbf{s}^{(i)}, \mathbf{s}^{(i)'})$, the reward function is

$$r^{\text{int}}(\mathbf{s}^{(i)}, \mathbf{s}^{(i)'}) \triangleq \log\left(c + \frac{1}{k}\sum_k R_k^{(i)}\right).$$

**Noise Contrastive Estimation.**   To train parameters $\psi$, CIC [28] minimizes the conditional entropy $\mathbb{H}[\mathbf{S}'; \mathbf{S} \mid \mathbf{Z}]$ using a noise contrastive estimation (NCE) [23, 37]. Denote $\mathbf{h} = f_\psi(\mathbf{s})$ and $\mathbf{h}' = f_\psi(\mathbf{s}')$ the projected representations of $\mathbf{s}$ and $\mathbf{s}'$, respectively. Furthermore, let $g_{\phi_z} : \mathbb{R}^d \to \mathbb{R}^d$ be an MLP that projects the skill into a latent representation and $g_{\phi_s} : \mathbb{R}^d \times \mathbb{R}^d \to \mathbb{R}^d$ be an MLP that projects $\mathbf{h}$ and $\mathbf{h}'$ into a latent representation. We define the NCE loss

$$\mathcal{L}_{\text{nce}}(\mathbf{s}^{(i)}, \mathbf{z}^{(i)}) = -\log \frac{\exp c_i^{(i)}}{\sum_{j=1}^{N} \exp c_j^{(i)}} \tag{5}$$

as an approximation to $\mathbb{H}[S'; S \mid Z]$, where we define $\cos(\cdot, \cdot)$ as the cosine similarity operator, $\omega$ as a temperature term, and $c_j^{(i)} = \cos\left(g_{\phi_z}(\mathbf{z}^{(i)}), g_{\phi_s}(\mathbf{h}^{(j)}, \mathbf{h}^{(j)'})\right)/\omega$. As aforementioned, this term serves as representation learning for parameters $\psi$. Intuitively, the NCE loss [23] pushes $p(\mathbf{s}' \mid \mathbf{s}, \mathbf{z})$ to be a delta-like density function. However, since we define behaviors with $n$ steps, we want a wider distribution, so in practice, $\omega$ is set to $0.5$ to smooth the distribution.

## 4   Mixture Of Suprises (MOSS)

Our goal is to provide an algorithm that does not build on any assumption about the entropy of the environment's dynamics. A straightforward way to achieve this is to maximize and minimize surprise simultaneously. A benefit of such an objective is that it promotes exploration to find a more effective policy for minimizing the surprise [6]. In particular, in [6], they maximize surprise according to all prior experiences while minimizing surprise according to the current episode. However, they find that it only helps in some cases. The challenge is to find an effective way to optimize these two opposite objectives since if we add the two objectives, it would simply yield a linear interpolation of the two objectives, which still corresponds to either maximizing *or* minimizing. A previous work [17], dubbed Adversarial Surprise (AS), resolves this challenge using an adversarial game where two policies compete against each other: one maximizes surprise while the other minimizes surprise. AS [17] requires training two policies that do not share parameters or data. Instead, we propose a novel yet simple mixture of policies to maximize and minimize surprise simultaneously. Our mixture of policies does not rely on an adversarial game [17] in which training is known to be challenging [35] and uses a single network for both objectives resulting in higher sample efficiency. Mainly, our method only

---

**Algorithm 1** MOSS: Unsupervised pretraining

---

**Input:** Environment, number of pretraining steps $N_T$.
Initialize DDPG, and $\phi_s, \phi_z, \psi$
**for** $t = 1$ **to** $N_T$ **do**
  **if** $(t\%\text{update skill every}) == 0$ **then**
    Sample $m_t \sim p(m)$            ▷ setting to $m = 0$ is equivalent to CIC[28]
    Sample $\mathbf{z}_t \sim p(\mathbf{z} \mid m_t)$
  **end if**
  Take some action $\mathbf{a}_t$ and observe $\mathbf{s}_{t+1}$
  Store $(\mathbf{s}_t, \mathbf{s}_{t+1}, m_t, \mathbf{z}_t)$ into buffer $\mathcal{B}$
  **if** $t \geq 4000$ **then**
    Sample a batch $\{(\mathbf{a}^{(i)}, \mathbf{s}^{(i)}, \mathbf{s}^{(i)'}, m^{(i)}, \mathbf{z}^{(i)})\}_{i=1}^N$ and compute intrinsic reward $r^{\text{int}}$ using equation 6.
    Update DDPG using intrinsic reward.
    Update $\phi_s, \phi_z, \psi$ using noise contrastive loss in equation 5.
  **end if**
**end for**

---

requires a minor change to CIC [28] that does not involve any additional computational cost during training. In particular, our mixture of policies builds on a competence-based approach [28], meaning we use two disjoint skill sets. We separate the two opposite objectives using a different skill set for each objective: one skill set for surprise maximization and another for surprise minimization.

Specifically, we define the mixture of policies using a binary random variable $M$, where we associate each mixture component with a continuous uniform distribution $\mathcal{U}^d(a, b)$ on an interval $[a, b]$. Each uniform distribution constitutes a skill distribution. Maintaining two different skill distributions allows us to define a different objective for each distribution. In particular, we define two distributions $\mathbf{Z}_{\max} \sim P(\mathbf{Z} \mid M = 0)$ and $\mathbf{Z}_{\min} \sim P(\mathbf{Z} \mid M = 1)$. Since we do not assume any priors on the entropy of the environment's dynamics, we deterministically set $M = 0$ for the first half of the steps and $M = 1$ for the other half of the steps in the episode. Compared to more sophisticated methods (*e.g.*, appendix B) for setting $M$ (*i.e.*, switching the objective), such a heuristic adds no computational cost during training, requires no hyper-parameter tuning, and performs well.

Moreover, motivated by the discussions in Section 3.2, we build our method on top of the CIC [4] [28] framework. This framework is appealing as it bypasses the chicken and egg problem and supports high-dimensional skills. In addition, it significantly outperforms previous competence-based methods on the URLB benchmark [29]. Hence, following CIC [28], we use the decomposition in Eq. (1)

$$\mathbb{I}(\mathbf{S}', \mathbf{S}; \mathbf{Z}) = \mathbb{H}[\mathbf{S}', \mathbf{S}] - \mathbb{H}[\mathbf{S}', \mathbf{S} \mid \mathbf{Z}].$$

In particular, we define *surprise* as the joint entropy, which corresponds to maximizing and minimizing $\mathbb{H}[\mathbf{S}', \mathbf{S}]$, and as in CIC [28] we use Eq. (4) to approximate the joint entropy. We approximate the conditional entropy $\mathbb{H}[\mathbf{S}', \mathbf{S} \mid \mathbf{Z}]$ using a noise contrastive estimation following Eq. (5). By minimizing the conditional entropy, we distill behaviors into skills. Doing so also serves as representation learning for computing the joint entropy [28].

**Implementation.** To ensure fairness, all hyper-parameters are kept as in CIC [28] and we use the same RL algorithm (*i.e.*, DDPG [31] as implemented in DrQ-V2 [51]).

In practice, $\mathbf{Z}_{\max} \sim P(\mathbf{Z} \mid M = 0) \triangleq \mathcal{U}^d(0, 1)$ and $\mathbf{Z}_{\min} \sim P(\mathbf{Z} \mid M = 1) \triangleq \mathcal{U}^d(-1, 0)$ (we refer readers to appendix for other variants that we tried), where we set $M = 0$ for the first half of steps and $M = 1$ for the remaining half of steps in the episode. Hence, the RL agent maximizes

$$\mathcal{F}(\theta) \triangleq -\mathbb{E}_{\mathbf{z}_{\max} \sim \mathcal{U}^d(0,1)} \mathbb{E}_{p_{\mathbf{z}_{\max}}(\mathbf{s}', \mathbf{s})}[\log p_{\mathbf{z}_{\max}}(\mathbf{s}', \mathbf{s})] + \mathbb{E}_{\mathbf{z}_{\min} \sim \mathcal{U}^d(-1,0)} \mathbb{E}_{p_{\mathbf{z}_{\min}}(\mathbf{s}', \mathbf{s})}[\log p_{\mathbf{z}_{\min}}(\mathbf{s}', \mathbf{s})].$$

---

[4] The implementation provided by CIC (https://github.com/rll-research/cic) is based on the arxiv version (https://arxiv.org/abs/2202.00161v2) which is an updated version of the ICLR version (https://openreview.net/forum?id=kOtkgUGAVTX). In particular, in the arxiv version, the intrinsic reward only relies on the KNN estimates and the NCE term is only used to train $\phi_s, \phi_z, \psi$. While, the ICLR version includes the NCE term in the intrinsic reward.

And the intrinsic reward is a function of the state-transition and the binary random variable $M$

$$r^{\text{int}}(\mathbf{s}^{(i)}, \mathbf{s}^{(i)'}, M) = \begin{cases} \log\left(c + \frac{1}{k}\sum_k R_k^{(i)}\right) & M = 0, \\ -\log\left(c + \frac{1}{k}\sum_k R_k^{(i)}\right) & M = 1. \end{cases} \tag{6}$$

As in CIC [28], we approximate the joint entropy using Eq. (4).

MOSS's implementation is straightforward; *i.e.*, at each time step, instead of sampling $\mathbf{Z} \sim P(\mathbf{Z})$, we sample $\mathbf{Z} \sim P(\mathbf{Z} \mid M)$ and replace the intrinsic reward with Eq. (6). Pseudocode for MOSS is provided in Alg. 1. In fact, fixing $M = 0$ yields CIC [28].

## 5   Experiments

**Environments.**   We present the main results of method by evaluating on the Unsupervised Reinforcement Learning Benchmark (URLB) [29], which is a standard unsupervised RL benchmark for continuous control. URLB [29] has three different domains listed in order of increasing difficulty:

- Walker is a biped robot domain constrained to a 2D vertical plane. The challenge for this domain is to learn how to balance and maneuver simultaneously.
- Quadruped is a four-legged robot in 3D space. This domain has a higher dimensional state and action space than the Walker domain.
- Jaco is a 6-DOF robotic arm domain with a three-finger gripper in 3D space that has to learn action constraints and manipulation.

Furthermore, each of the three domains has four different downstream tasks with varying difficulties. We refer readers to Tab. 1 for the complete list of 12 tasks. Finally, in OpenAI gym environments [8], since an episode terminates as soon as a robot falls, the agent will naturally avoid falling to the ground as it will get a 0 reward. Hence, during pretraining, the OpenAI gym environment [8] may leak some task information (*e.g.*, standing) to the unsupervised learning agent [28]. Therefore, URLB [29] builds on top of the Deepmind Control Suite [48] to avoid leaking some task information.

The domains in URLB [29] use low-dimensional state information, deterministic transitions, and non-evolving environments. In contrast, ViZDoom [27] uses raw pixel observations, stochastic transition, and evolving environments. Furthermore, since the environment randomly spawns enemies that shoot fireballs, the agent faces external perturbations that surprise the agent. We evaluate our method on the *DefendTheLine* and *TakeCover* maps. Please refer to Figure 2 for environment renders.

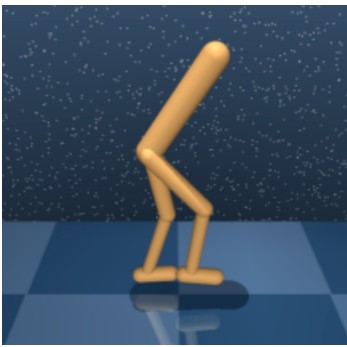
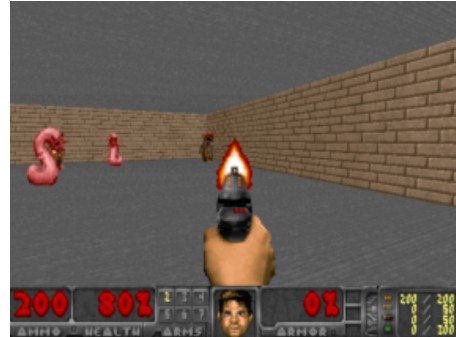

|  (a) URLB based on DMC  |  (b) The ViZDoom Environment.  |

Figure 2: **Evaluation Environments.** We evaluate our method in (a) URLB, a standard unsupervised reinforcement learning benchmark based on continuous control (b) ViZDoom a pixel observation based reinforcement learning environment based on the Doom game with multiple maps.

**Evaluation.**   We follow the benchmark's standard training procedure by pretraining the agent for 2 million steps in each of the three domains and then finetuning the pre-trained agent for $100k$ steps with downstream task rewards. Since AS trains two network-independent policies, we ensure both policies each perform 2 million environment steps during pretraining. To ensure fairness, we pretrain

and finetune each method with the same 12 seeds (0-11) using the code provided in URLB[5] [29]. We present the results in Fig. 3 for a total of $1584$ ($= 11$ algorithms $\times 12$ tasks $\times 12$ seeds) runs. Our main evaluation metrics are interquartile mean (IQM) and Optimality Gap (OG) of normalized scores, where we used the score of a DDPG [51] agent trained from scratch for 2 million steps as the expert scores [29]. On the one hand, IQM is more robust to outliers than the mean, and IQM is significantly less biased than the median [2]. On the other hand, the Optimality Gap measures how far scores are from the expert scores [2].

**Baselines.** Baselines include methods presented in the URLB benchmark [29] as well as CIC [28]. The baseline methods fall into knowledge-based methods, *i.e.*, ICM [38], Disagreement [39], RND [10], data-based methods, *i.e.*, APT [33], Proto-RL [52], AS [17], and competence-based methods, *i.e.*, SMM [30], DIAYN [15], APS [32], CIC [28]. We refer readers to the Appendix of URLB [29] for details on the different baselines. Lastly, since the two policies (Alice and Bob) in AS are parameter-independent, we pretrain for 4 million steps, each policy trained using 2 million steps.

We evaluate our agent in ViZDoom [27] against CIC [28], and a modified version of CIC [28], dubbed NegativeCIC, where the intrinsic reward is the negative joint entropy. Since the ViZDoom environment has high natural environment dynamics entropy, NegativeCIC should act as a strong baseline. We adopt the same pretraining and finetuning procedure as in URBL for ViZDoom.

**MOSS.** We keep all hyperparameters as those in CIC [28]. Specifically, unless specified otherwise, we set $M = 0$ for the first half of steps and $M = 1$ for the other half of steps in the episode. We refer readers to the Appendix for more details.

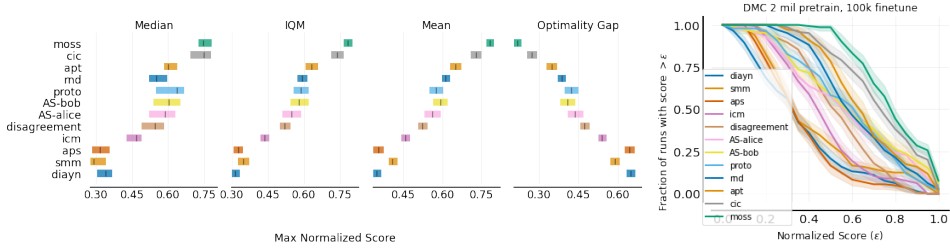

(a) **Aggregate Statistics**. MOSS obtains the highest IQM and the lowest Optimality Gap.

(b) **Performance Profiles.** MOSS stochastically dominates all other methods.

Figure 3: **Main results.** We report aggregate statistics (a) and performance profiles (b) following [2] for 12 downstream tasks on URLB with 12 seeds, providing a total of 144 seeds.

## 5.1 Results

Table 1: **Numerical results.** Following URLB [29] we show the mean and standard error over 12 seeds (0-11). We denote competence-based methods, knowledge-based, and data-based methods in blue, black, and red, respectively.

| Domain | | Walker | | | | Quadruped | | | | Jaco | | |
|---|---|---|---|---|---|---|---|---|---|---|---|---|
| Task | Flip | Run | Stand | Walk | Jump | Run | Stand | Walk | Bottom Left | Bottom Right | Top Left | Top Right |
| ICM[38] | 381±10 | 180±15 | 868±30 | 568±38 | 337±18 | 221±14 | 452±15 | 234±18 | 112±7 | 94±5 | 90±6 | 93±11 |
| Disagreement [39] | 313±8 | 166±9 | 658±33 | 453±37 | 512±14 | 395±12 | 686±30 | 358±25 | 120±7 | 132±5 | 111±10 | 113±10 |
| RND [10] | 412±18 | 267±18 | 842±19 | 694±26 | **681±11** | 455±7 | 875±25 | 581±42 | 106±6 | 111±6 | 83±7 | 107±5 |
| ICM APT[33] | 596±24 | 491±18 | 949±3 | 850±22 | 508±44 | 390±24 | 676±44 | 464±52 | 114±5 | 120±3 | 116±4 | 114±8 |
| Proto [52] | 378±4 | 225±16 | 828±24 | 610±40 | 426±32 | 310±22 | 702±59 | 348±55 | 130±12 | 131±11 | 134±12 | 146±10 |
| AS-Bob | 475±16 | 247±23 | 917±36 | 675±21 | 449±24 | 285±23 | 594±37 | 353±39 | 116±21 | **166±12** | 143±12 | 139±13 |
| AS-Alice | 491±20 | 211±9 | 868±47 | 655±36 | 415±20 | 296±18 | 590±41 | 337±17 | 109±20 | 141±19 | 140±17 | 129±15 |
| SMM[30] | 428±8 | 345±31 | 924±9 | 731±43 | 271±35 | 222±23 | 388±51 | 167±20 | 52±5 | 55±2 | 53±2 | 57±5 |
| DIAYN [15] | 306±12 | 146±7 | 631±46 | 394±22 | 491±38 | 325±21 | 662±38 | 273±19 | 35±5 | 35±6 | 23±3 | 31±5 |
| APS [32] | 355±18 | 166±15 | 667±56 | 500±40 | 283±22 | 206±16 | 379±31 | 192±17 | 61±6 | 79±12 | 51±5 | 56±7 |
| CIC [28] | 715±40 | **535±25** | **968±2** | 914±12 | 541±31 | 376±19 | 717±46 | 460±36 | 147±8 | 150±6 | 145±9 | 139±9 |
| MOSS (Ours) | **729±40** | 531±20 | 962±3 | **942±5** | 674±11 | **485±6** | **911±11** | **635±36** | **151±5** | 150±5 | **150±5** | **150±6** |

---

[5]Our experiments are based on the code of URLB `https://github.com/rll-research/url_benchmark`, and CIC `https://github.com/rll-research/cic`

To ensure the reliability of our main results, we report results as described in [2] to account for the uncertainty. We present results in Fig. 3, where, surprisingly, our method obtains state-of-the-art performance on the URLB benchmark [29] on our main evaluation metrics [2]. In particular, the performance profile in Fig. 3(b) reveals that MOSS stochastically dominates all other methods. We also present numerical results in Tab. 1 for a detailed breakdown over individual tasks.

For the IQM metric, we report an increase of $6\%$ compared to CIC, which is the second leading method. Furthermore, as for the Optimality Gap metric, we decreased $20\%$ compared to CIC. Furthermore, in Fig. 3(b) previous methods intersect at multiple points (making it hard to conclude whether a method is better than the other), while MOSS is always above all previous methods. In addition, in Fig. 3(b), MOSS maintains all runs greater than a normalized score as high as $0.5$.

Table 2: **ViZDoom Results**. We report on two maps from ViZDoom to illustrate the robustness of our method in different environments.

|  | Defend the Line | Take Cover |
| --- | --- | --- |
| NegativeCIC | 462±11 | 51793±2936 |
| MOSS | **510±22** | **63497±2384** |

We also evaluated our method on ViZDoom [27]. Since the *TakeCover* and *DefendTheLine* maps in ViZDoom provides natural perturbations, we use NegativeCIC as the baseline. We present the numerical results in Tab. 2. Since the environments in ViZDoom are highly stochastic, we remove the top and bottom 10% of scores and report the mean and standard error to account for any outliers. Like in the main results presented earlier, MOSS consistently outperforms the baseline method under this completely new setting.

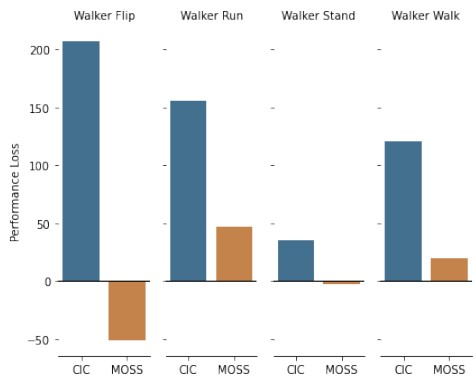

(a) **Ablation on environment.** Adding stochasticity in the URLB environment hurts CIC. We notice that MOSS is significantly more robust to the stochasticity than CIC.

(b) **Ablation on policy.** Removing the mixture significantly hurts the performance.

Figure 4: **MOSS ablation experiments** on the Walker domain [29]. In (a), we inject stochasticity by adding a Gaussian noise on the agent's action according to a Bernoulli with $p = 0.3$ and report performance loss. In (b), we report performance for CIC, MOSS_SAME, and NegativeCIC. Specifically, MOSS_SAME represents MOSS without the mixture of policies, and NegativeCIC represents CIC with the intrinsic reward as the negative joint entropy.

## 5.2 Ablation

Since MOSS does not make assumptions about the entropy of the environment's dynamics, we are interested in knowing whether adding stochasticity affects performance. For this stochastic setup, during each interaction with the environment, we sample $Y \sim \text{Bernoulli}(0.3)$ and if $Y = 1$, we add a Gaussian Noise $\mathcal{N}(0, 0.2)$ to the agent's intended action. In Fig. 4(a), we observe that CIC's [28] performance drops while MOSS' performance is almost unaffected. This finding empirically supports previous works which claim that solely maximizing surprise for unsupervised RL agents is sometimes not enough, especially in stochastic environments [6, 41].

In MOSS, we maintain two different skill distributions, $i.e.$, $\mathbf{Z}_{\max} \sim \mathcal{U}^d(0, 1)$ and $\mathbf{Z}_{\min} \sim \mathcal{U}^d(-1, 0)$. In particular, setting $\mathbf{Z}_{\min}$ to the same distribution as $\mathbf{Z}_{\max} \sim \mathcal{U}^d(0, 1)$ is equivalent to directly optimizing CIC [28] with the two objectives simultaneously. We dub MOSS with the same skill distribution, MOSS_SAME. In Fig. 4(b), we observe that naively doing so can result in poor performance, showing the effectiveness of our method in mitigating the simultaneous optimization of two contradicting objectives.

Moreover, as shown in Fig. 4(b), NegativeCIC performs poorly on the URLB benchmark [29] since most tasks required structured exploration in a low entropy environment.

## 6 Discussions & Limitations

We presented MOSS, an unsupervised RL method that does not make assumptions about the environment dynamics' entropy by simultaneously maximizing and minimizing surprises. Our method falls into competence-based methods. Leveraging a mixture of policies for more than surprise maximization and minimization is an interesting future research.

**Limitations.** One limitation is that our method has a simple and effective deterministic rule for switching between the two objectives *independent* of state or history. We did explore a more sophisticated method in Appendix B that had a better performance but required tuning and extra computation. Therefore, a promising direction is to explore methods that take advantage of state information (*e.g.*, using a meta-controller [26]) to output a more optimal switching mechanism that incentivizes deep structured exploration [40]. Moreover, we only considered competence-based methods. Extending our method to other unsupervised RL methods would be interesting to investigate in future work. Additionally, when finetuning, we transferred network weights which could lead to unlearning of already learned behaviors [11]. It would be meaningful to combine with works that circumvent this issue like [11] which utilized frozen pretrained policies to improve downstream finetuning further. Lastly, we only considered two objectives, and including more meaningful objectives like empowerment or anxiety [36, 46] during pretraining could be an exciting direction.

**Potential Negative Impact.** Since we operate under the unsupervised reinforcement learning domain, the agent receives rewards that incentives exploration. Therefore, agents under this domain can be more unpredictable than agents that receive task rewards for intended behaviors. Furthermore, there is no current safeguard for the agent to not act maliciously in its environment; the agent may potentially cause harm to properties situated in the environment, including the agent itself. Therefore, unsupervised learning with safety constraints would be an exciting area of future research to mitigate this potential negative impact.

**Acknowledgement.** This work is supported in part by the National Science and Technology Major Project of the Ministry of Science and Technology of China under Grants 2018AAA0101604, the National Natural Science Foundation of China under Grants 62022048 and the State Key Lab of Autonomous Intelligent Unmanned Systems. We also appreciate the generous donation of computing resources by High-Flyer AI.

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
