# A  Additional Implementation Details

Table 1: **Hyperparameters for MOSS and DDPG.** These hyperparameters are fixed throughout all domains.

| DDPG hyper-parameter | Value |
|---|---|
| Replay buffer capacity | $10^6$ |
| Action repeat | 1 |
| Seed frames | 4000 |
| n-step returns | 3 |
| Mini-batch size | 1048 |
| Discount ($\gamma$) | 0.99 |
| Optimizer | Adam |
| Learning rate | $10^{-4}$ |
| Agent update frequency | 2 |
| Critic target EMA rate ($\tau_Q$) | 0.01 |
| Features dim. | 1024 |
| Hidden dim. | 1024 |
| Exploration stddev clip | 0.3 |
| Exploration stddev value | 0.2 |
| Number pre-training frames | $2 \times 10^6$ |
| Number fine-turning frames | $1 \times 10^5$ |

| MOSS hyper-parameter | Value |
|---|---|
| Skill dim | 64 |
| $\mathbf{Z}_{max}$ Prior | Uniform[0,1] |
| $\mathbf{Z}_{min}$ Prior | Uniform[-1,0] |
| Update skill frequency | 50 |
| State net $f_\psi$ | $\dim \mathcal{O} \to 1024 \to 1024 \to 64$ ReLU MLP |
| Skill net $g_{\phi_z}$ | $64 \to 1024 \to 1024 \to 64$ ReLU MLP |
| Prediction net $g_{\phi_s}$ | $64 \to 1024 \to 1024 \to 64$ ReLU MLP |
| Episode partition | 2 |

## A.1  MOSS Implementation

We implement MOSS using JAX [1], Haiku [4]. We chose to build on top of JAX [1] as we observed a 2x speedup compared to PyTorch [8]. Moreover, we use PyTorch [8] and Reverb [2] to implement the replay buffer. Tab. 1 details the hyper-parameters used in MOSS which are taken directly from CIC [6]. Since MOSS builds on CIC [6], we empirically verified that our implementation on top of JAX [1] matches the performance on top of PyTorch [8].

Training follows the URLB benchmark [7], where an agent is pretrained for 2 million steps and then finetuned on a downstream task for $100k$ steps.

## A.2  Baseline Implementation

For the baselines that were presented in URLB, we obtained the results by running the code from the URLB GitHub repo: `https://github.com/rll-research/url_benchmark`. All hyperparameters were kept the same as the original implementation.

## A.3  Environment

For continuous control domains, we used the custom Deep Mind Control Suite [10] environments from the URL Benchmark GitHub repo: `https://github.com/rll-research/url_benchmark`. For the VizDoom domain, we used code from their official GitHub repo: `https://github.com/mwydmuch/ViZDoom`. We include the environment renders in Figure **??**.

Table 2: **Hyperparameters for MOSS and DQN.** These hyperparameters are fixed throughout all domains.

| Double-DQN hyper-parameter | Value |
| --- | --- |
| Replay buffer capacity | $10^6$ |
| Action repeat | 1 |
| Frame repeat | 12 |
| Seed frames | 4000 |
| n-step returns | 3 |
| Mini-batch size | 1048 |
| Discount ($\gamma$) | 0.99 |
| Optimizer | Adam |
| Learning rate | 0.0001 |
| Agent update frequency | 2 |
| Critic target EMA rate ($\tau_Q$) | 0.01 |
| Hidden dim. | 256 |
| Epsilon Schedule ($\epsilon$) | pretrain: *linear_decay*(start=1.0, end=0.1, steps=$10^5$) 
 finetune: *linear_decay*(start=1.0, end=0.1, steps=$10^4$) |
| Number pre-training frames | $10^6$ |
| Number fine-turning frames | $10^5$ |

| MOSS hyper-parameter | Value |
| --- | --- |
| Skill dim | 64 |
| Num skills | 80 |
| Prior | Discrete-Uniform |
| Update skill frequency | 50 |
| Episode partition | 5 |

## A.4 Double DQN

We made modifications to MOSS to evaluate in discrete action settings. Specifically, we adopted Double-DQN [11] as the backbone reinforcement learning algorithm, which has the Q-value target as

$$Q_t^{\text{target}} = R_{t+1} + \gamma Q(s_{t+1}, \underset{a}{\text{argmax}} Q(s_{t+1}, a; \theta), \theta_t^-),$$

where $\theta_t, \theta_t^-$ are online network parameters and the target network parameters at time $t$, respectively.

Furthermore, since ViZDoom [5] has a smaller action space, we provided the agent with a discrete number of skill embeddings, similar to DIAYN [3]. Moreover, since performing CPC loss [6] in high dimensional pixel space is not ideal, and to save the number of parameters, we use the CNN backbone of the agent's DQN network to project observations into state vectors, then use an MLP identical to the continuous action setting to calculate the CPC loss [6] using the discrete set of skill embeddings. Tab. 2 details the hyper-parameters used for Double DQN and MOSS in the ViZDoom environment.

## A.5 Compute Resources

All experiments were run on an internal cluster with 8 NVIDIA A100 GPU and AMD EPYC 7742 64-Core Processor. Pretraining MOSS takes roughly 5 hours. While finetuning takes roughly 30 mins.

## A.6 Thing that did not work: Skill Distribution

In MOSS, we maintain two different skill distributions, *i.e.*, $\mathbf{Z}_{\max} \sim P(\mathbf{Z} \mid M = 0)$ and $\mathbf{Z}_{\min} \sim P(\mathbf{Z} \mid M = 1)$. In particular, we defined them as $\mathbf{Z}_{\max} \sim \mathcal{U}^d(0, 1)$ and $\mathbf{Z}_{\min} \sim \mathcal{U}^d(-1, 0)$. Given a skill of dimension $d$, another way to define $P(\mathbf{Z} \mid M = 0)$ and $P(\mathbf{Z} \mid M = 1)$ is to allocate half of a skill vector for $m = 0$ and the other half for $m = 1$. In other words, $\mathbf{Z}_{\max}$ and $\mathbf{Z}_{\min}$ are both drawn from $\mathcal{U}^d(0, 1)$, however, when $m = 0$, $(\mathbf{Z}_{\max})_{:d/2} \sim P(\mathbf{Z} \mid M = 0)$ while $(\mathbf{Z}_{\max})_{d/2:} = \mathbf{0}$. Conversely, when $m = 1$, $(\mathbf{Z}_{\min})_{d/2:} \sim P(\mathbf{Z} \mid M = 0)$ while $(\mathbf{Z}_{\max})_{:d/2} = \mathbf{0}$.

# B  Objective Switching

Table 3: **Adaptive Mode Switching Results on Quadruped**. We report the results of adaptive mode switching MOSS on the quadruped domain with $\beta = 1.1$ obtained from grid search

| Task | MOSS | MOSS$^{\text{adaptive}}$ |
|---|---|---|
| Quadruped Jump | 674±11 | **687±36** |
| Quadruped Run | 485±6 | **512±20** |
| Quadruped Stand | **911±11** | 869±26 |
| Quadruped Walk | 635±36 | **758±37** |

Since our framework has two objectives, the reinforcement learning agent requires collecting experience to train its conditional policy under both proposed objectives. Moreover, the temporal structure of staying in different modes of behavior in humans and animals is not monolithic; therefore, designing when to switch modes for the agent might be an interesting area to consider. For example, previous works investigated a similar setting and used a threshold based on Q-values [9]. However, in unsupervised reinforcement learning, without any sense of the possible downstream task distribution, we wish to design a mode switching mechanism that is both adaptive and scale-invariant across environments.

Literature in active learning proposes a strategy to query high uncertainty samples. We adopt this framework to MOSS. Intuitively, we wish to encourage the agent to collect more trajectories at places with higher *epistemic uncertainty* for the agent to learn more in a more unfamiliar situation. Since we do not wish to train additional networks, we use the online critic network to output a sample variance to act as a proxy for uncertainty. Concretely, for a given mode $m$, we sample a batch $N$ of skill-vectors $z_m^{(i)} \sim p(z|m)$. We then calculate the sample variance as

$$\mathbb{V}_{t,m}(Q_m^\pi) = \frac{\sum_{i=1}^{N} (Q(s_t, \pi(\cdot|s_t, z_m^{(i)}), z_m^{(i)}) - \bar{Q}_m^\pi)^2}{N-1},$$

where $\bar{Q}_m^\pi$ is the sample mean Q-value for the batch.

However, since variance is a measurement with units, and we do not wish to introduce additional environment-specific hyperparameters, we use the history of variance and the current Q-value variance as function inputs to select modes, bypassing the unit sensitivity problem across environments. Concretely, we keep a record of the sample variance of both modes at each time step $t$, denoted as $\mathbb{V}_{t,M=0}$ for maximization skills and $\mathbb{V}_{t,M=1}$ for minimization skills. At each skill switching interval, the agent selects the mode $m_t$ for the current time step by,

$$m_t = \begin{cases} -(m_{t-k} - 1) & \beta * \mathbb{V}_{t-k,-(m_{t-k}-1)} \leq \mathbb{V}_{t,-(m_{t-k}-1)} \\ m_{t-k} & \text{otherwise}, \end{cases}$$

where $\beta$ is a coefficient greater than 1 that controls the threshold of switching modes, and $k$ is the interval we use to switch skills and modes. Intuitively, we wish to switch to the opposite mode if the opposite mode's uncertainty at time $t$ is greater than a coefficient times the last recorded uncertainty. We show some preliminary results for the quadruped domain in Tab. 3. The effectiveness of adaptive methods is demonstrated in the results, where MOSS$^{\text{adaptive}}$ was able to beat out MOSS. However, this method requires memory and computational resources and we found that the hyperparameter $\beta$ is somewhat sensitive across domains and requires tuning. Therefore, we leave additional investigations for mode switching to future work because presenting an efficient and hyperparameter-insensitive method of maximization and minimization of surprise was the main scope of this paper.

# C  Additional Results

## C.1  Prior Distribution $P(M)$

In MOSS, we deterministically set $M = 0$ for the first half of the steps and $M = 1$ for the other half of the steps in the episode. This corresponds to having $50\%$ of maximization data and $50\%$ of minimization data. We report additional results on Walker for different ratios of maximization and minimization in Tab. 4.

Table 4: **Ablation on prior** $M$. A prior towards entropy maximization results in a better performance than a prior towards entropy minimization.

|  | Walker Flip | Run | Stand | Walk |
|---|---|---|---|---|
| 30% maximization | 723±36 | 515±32 | 967±3 | 921±18 |
| 40% | 738 ±42 | 526±15 | 968±2 | 908±24 |
| 50% (MOSS) | 729±40 | 531±20 | 962±3 | 942±5 |
| 60% | 781±38 | 599±11 | 967±1 | 935±6 |
| 70% | 785±39 | 567±15 | 965±3 | 933±7 |

We observe that a prior towards entropy maximization results in a better performance than a prior towards entropy minimization on Walker.

Table 5: **Ablation on prior** $Z$. A continuous skill performs better than a discrete skill but not by a large margin. In addition, a higher dimensional skill tends to perform better.

|  | Continuous 1 | Continuous 16 | Continuous 64 (MOSS) | Discrete 1 |
|---|---|---|---|---|
| Walker Flip | 827±50 | 934±12 | 729±40 | 672±9 |
| Walker Run | 329±41 | 245±44 | 531±20 | 335±20 |
| Walker Stand | 930±7 | 830±25 | 962±3 | 931±8 |
| Walker Walk | 954±4 | 960±2 | 942±5 | 751±70 |
| Quadruped Jump | 193±27 | 649±25 | 674±11 | 194±50 |
| Quadruped Run | 167±17 | 444±26 | 485±6 | 162±20 |
| Quadruped Stand | 306±32 | 834±33 | 911±11 | 267±38 |
| Quadruped Walk | 158±19 | 569±49 | 635±36 | 217±23 |

## C.2 Skill Prior

We present additional results on the dimensionality of the skill vector along with the event space (discrete vs continuous) in Tab. 5. We find that a continuous skill performs better than a discrete skill but not by a large margin. However, it seems that in general, a higher dimensional skill performs better, especially in Quadruped which has a higher dimensional state and action space than the Walker domain. Our results suggest that the optimal skill vector may be task-dependent.

Table 6: **Zero-shot results.** The zero-shot performance of methods on the URL Benchmark

| Domain | Walker | | | | Quadruped | | | | Jaco | | | |
|---|---|---|---|---|---|---|---|---|---|---|---|---|
| Task | Flip | Run | Stand | Walk | Jump | Run | Stand | Walk | Reach Bottom Left | Reach Bottom Right | Reach Top Left | Reach Top Right |
| ICM | 78±3 | 32±1 | 170±6 | 60±4 | 225±33 | 150±22 | 299±44 | 153±23 | **9±2** | **7±2** | 21±3 | **10±3** |
| Disagreement | 207±2 | 78±0 | 366±3 | 167±2 | 464±10 | 269±6 | 534±11 | 273±7 | 5±1 | 3±1 | **27±4** | 9±2 |
| RND | **237±3** | 89±1 | 392±4 | 195±3 | 549±7 | 319±4 | 638±6 | 321±9 | 4±1 | 3±1 | 17±2 | 4±0 |
| APT | 235±3 | **91±0** | **401±3** | **204±3** | 304±38 | 194±22 | 384±44 | 198±24 | 1±1 | 0±0 | 0±0 | 0±0 |
| Proto | 171±4 | 72±2 | 322±9 | 162±6 | 36±5 | 23±3 | 46±6 | 23±3 | 1±0 | 1±0 | 10±2 | 4±1 |
| AS-BOB | 35±3 | 22±0 | 132±4 | 28±2 | 170±37 | 111±24 | 223±49 | 107±21 | 1±1 | 0±0 | 1±1 | 0±0 |
| AS-ALICE | 31±2 | 24±1 | 139±9 | 28±2 | 195±26 | 131±17 | 264±34 | 127±16 | 0±0 | 0±0 | 0±0 | 0±0 |
| SMM | 117±11 | 49±4 | 229±13 | 100±16 | 73±23 | 46±14 | 91±27 | 48±15 | 1±0 | 4±2 | 9±2 | 6±1 |
| DIAYN | 22±2 | 18±1 | 107±9 | 20±2 | 136±31 | 91±20 | 181±41 | 93±21 | 1±0 | 2±1 | 3±1 | 4±1 |
| APS | 35±2 | 27±1 | 162±11 | 30±2 | 134±20 | 89±13 | 180±26 | 89±13 | 0±0 | 0±0 | 0±0 | 0±0 |
| CIC | 218±5 | 79±1 | 356±6 | 174±4 | 415±34 | 248±21 | 493±41 | 250±23 | 0±0 | 0±0 | 0±0 | 1±0 |
| MOSS | 166±5 | 58±1 | 295±9 | 112±3 | **627±28** | **370±16** | **767±35** | **306±12** | 0±0 | 2±1 | 0±0 | 1±0 |

## C.3 Zero-shot Results

We show zero-shot results of different URL methods in Tab. 6. In addition, Tab. 7 provides zero-shot results for both $\mathbf{Z}_{\max}$ and $\mathbf{Z}_{\min}$.

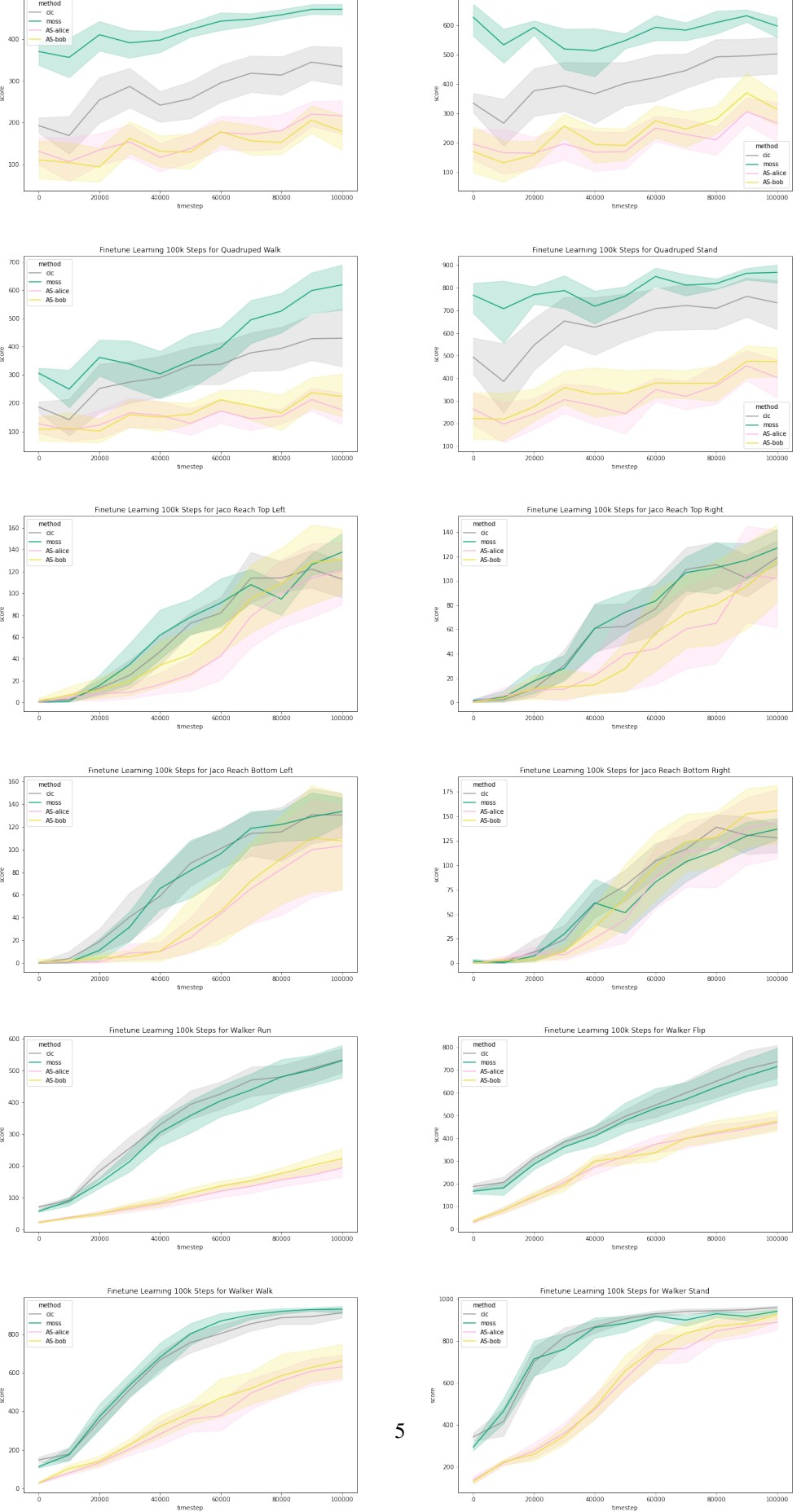

Table 7: **Zero-shot results of $Z_{\max}$ and $Z_{\min}$.** The comparison of zero-shot results of the two mixture of skills evaluated on the Quadruped domain.

| | Quadruped Jump | Quadruped run | Quadruped Stand | Quadruped Walk |
|---|---|---|---|---|
| $\text{MOSS}_{\text{min,frozen}}$ | 85.3±9.5 | 45±4 | 110±13 | 43±4 |
| $\text{MOSS}_{\text{max,frozen}}$ | 627±28 | 370±16 | 767±35 | 306±12 |

## C.4 Finetune Learning Curves

We include the finetune learning curves in Figure 1 for MOSS and two methods that are most related to MOSS, namely, CIC and Adversarial Surprise.

We can see that these methods generally improved with finetuning with task reward. However, we observe that the Quadruped domain had a noticeably flatter learning curve, meaning that methods benefited less from training than in other domains.