# OpenReview forum: "A Mixture Of Surprises for Unsupervised Reinforcement Learning"
_NeurIPS.cc/2022/Conference — NeurIPS 2022 Accept_

### Official Review · Reviewer_w2pW · 2022-07-10

**Rating:** 6
**Confidence:** 4
**Soundness:** 3 good
**Presentation:** 3 good
**Contribution:** 3 good

**Summary:**

This paper proposes an unsupervised RL algorithm for pretraining a policy. The algorithm is motivated by learning to both maximize and minimize the mutual information between latent skill variables and the state-transition joint distribution. The method approximates this mutual information following prior work, and proposes a learning algorithm that simply splits up the pretraining episodes into two phases: the first phase with intrinsic rewards corresponding to maximization, and the second phase corresponding to minimization. Experiments on the environments and tasks from the benchmark of [28] illustrate that this method achieves superior finetuning results compared to a variety of other methods. Ablations illustrate that this method has less degradation in performance than a maximizing phase alone (corresponding to prior work) when additional stochasticity is added to the transition dynamics, and an ablation that illustrates the importance of having the latent skill priors be different.

**Questions:**

- Please address the weaknesses:
  - Can you fix the terminology about the three types of methods?
  - Can you fix the characterization of the objective function?
  - Can you fix the ambiguity in the connection between the objective function and the optimization procedure?
  - Can you include the two baseline evaluations I requested?
  - Can you include the ablations I requested?
  - Can you include an empirical comparison to [18]?
- L170: "Intuitively, the NCE loss [ 23 ] pushes p(s′ | s, z) to be a delta-like density function. However, since we define behaviors with n steps, we want a wider distribution" The second sentence does not follow from the first. Even if p(s'|s,z) is deterministic, it can still be applied for N steps, to yield an N-step behavior. Can the paper clarify these statements?
- Can you explain the following inconsistency? L260 claims that setting Z_min = Z_max is equivalent to CIC, yet this results in different empircal performances in Fig 4b. By L260, we should expect their results to be equivalent. Why are they not? Or perhaps it's not equivalent to CIC, because it involves flipping the sign of the intrinsic reward function?

**Limitations:**

Yes.

**Strengths And Weaknesses:**

# Strengths
- The area of investigation is relevant to the community. Pretraining / unsupervised learning for RL agents is an active area of research interest.
- The proposed method is technically sound.
- The proposed method is technically straightforward.
- The proposed method demonstrates compelling performance compared to prior work.
- The experimental evaluation uses compelling aggregate statistics
- The experimental evaluation compares to a large variety of prior methods

# Weaknesses
- There is some undefined terminology. "Competence"-, "Data"-, and "Knowledge"-based methods were never defined. This makes some parts hard to understand, e.g. L99 "To motivate the potential of competence-based methods over data-based or knowledge-based methods". I recognize that these terms come from CIC, but this paper should either restate the definitions or refer to CIC.
- I think the objective function is mischaracterized. L183 "In particular, we define surprise as the state-transition entropy, which corresponds to maximizing and minimizing $H[S', S]$". Unless I misunderstand the notation, this is the entropy of the *joint* distribution $p(s',s)$, which is *not* the entropy of the state-transition dynamics distribution $p(s'|s,a)$, nor the entropy of the state-transition dynamics induced by the policy $p(s'|s) = E_\pi p(s'|s,a)$ (which are the only two distributions that I would refer to as being "state-transitions"). Therefore, I would not say that the objective is the state-transition entropy. It is *related* to the state transition entropy, since $H(S',S) = H(S'|S) + H(S)$, i.e. it is the entropy of the induced Markov dynamics plus another entropy. The problem with referring to this as only the "state-transition" entropy is that it fails to capture the fact that the value of H(S) affects the resulting value, so I think it is a mischaracterization of the objective. I think it is important to be very precise about this naming, because, in general, optimizing $H(S'|S)$ will lead to very different results than optimizing $H(S'|S)+H(S)$.
- The connection between the original objective and the resulting optimization procedure is unclear. The objective is $I((S', S) ; Z) = H((S', S)) - H((S', S) | Z)$. The CIC paper, upon which Alg 1 is based, uses both a noise-contrastive estimator (for the latter term) and a KNN estimator (for the former term). Alg 1 states the intrinsic reward is computed using Eq. 9, but this seems to only involve the noise constrastive estimator. Is Alg 1 correct? Where is the KNN estimator?
- The experiments do not adequately illustrate the effect that finetuning has on the pretrained policy. It is possible that the pretrained policy does not benefit from finetuning in some cases, e.g. taking $z \sim Z_{min}$ might lead to a high score on environments where the downstream tasks is very related to entropy minimization. Can you add to the evaluations two baselines for non-finetuned policies constructed from each of the $Z$ distributions (i.e. one for $\pi(a|s, z\sim p(Z_{max}))$, another for $\pi(a|s, z\sim p(Z_{min})$)? These should not be finetuned, instead, simply illustrate how related the learned policy is to the downstream task. I would call these something like $MOSS_{max,frozen}$ and $MOSS_{min,frozen}$ . Note that this doesn't require any additional learning, just an evaluation of these existing policies. Beyond this, you could also include the finetuning learning curves in the appendix, although that's a bit less important.
- The current ablation does not illustrate the importance of the choice of the structure of the Z priors, which includes both the dimensionality $d$, as well as the event space. One thing that seems most natural is simply to take z \in {0,1}, where 0 defines 'minimizing' and 1 defines 'maximizing', i.e. use $P_{min}(Z) \doteq \delta(z=0)$, and $P_{max}(Z)\doteq \delta(z=1)$. It is not clear what the additional dimensionality of $Z$ is useful for (the appendix tells us its 64 dimensional). Some discussion and empirical are needed to justify this choice. It could be the case that very low dimensional Zs are still useful in some (or all) tasks, intuitively, these tasks might be those in which there is only one 'useful' way each to minimize or maximize entropy, rather than environments in which repertoires (>=2 each) of entropy maximizing and minimizing skills is useful. Defining the skill variables to be continuous essentially admits learning an infinite number of skills, but it is not clear that this is the best choice.
- Although this paper pointed out the relevance, it did not include an empirical comparison to [18], which is also motivated by the task of learning to both maximize and minimize rewards. The inclusion of this comparison would significantly strengthen the empirical results.

# Minor weaknesses
- Please add the main method's scores to Figure 4(b).
- By the paper's definition of surprise as the entropy of the joint distribution of state transitions, the method is not optimizing *only* surprise (since it's just one of the two terms of the mutual information). The method name then becomes something of a misnomer...

---

> ### Author Response · Authors · 2022-08-02
> **Response to Reviewer w2pW (part 2/2)**
>
> **"Can you include an empirical comparison to [18]?"** \
> Thanks for this suggestion. We added a comparison with Adversarial Surprise (AS) [18] to Section 5 of the paper. We also list the numerical results in the table below:
> ||Walker Flip|Walker Run|Walker Stand|Walker Walk|Quadruped Jump|Quadruped Run|Quadruped Stand|Quadruped Walk|Jaco Bottom Left|Jaco Bottom Right|Jaco Top Left|Jaco Top Right|
> |:-|:-|:-|:-|:-|:-|:-|:-|:-|:-|:-|:-|:-|
> |MOSS|729±40|531±20|962±3|942±5|674±11|485±6|911±11|635±36|151±5|150±5|150±5|150±6|
> |AS-ALICE-4 MIL|491±20|211±9|868±47|655±36|415±20|296±18|590±41|337±17|109±20|141±19|140±17|129±15|
> |AS-ALICE-2MIL |469±13|195±14|914±17|655±33|380±18|279±9|543±21|276±19|114±20|158±15|143±13|120±20|
> |AS-BOB-4MIL|475±16|247±23|917±36|675±21|449±24|285±23|594±37|353±39|116±21|166±12|143±12|139±13|
> |AS-BOB-2MIL|474±23|223±15|937±15|687±42|427±20|262±8|590±22|314±23|118±21|168±15|154±13|121±14|
>
> Since Alice and Bob do not share parameters, we show finetuning results pretrained using 4 million environment steps, splitting 2 million to each Alice and Bob. We notice a slight increase in performance with longer pretraining for AS. Results show that AS is a competitive approach in the URL setting. In addition, AS obtained a noticeably high performance in the Jaco domain. But, overall, MOSS outperforms AS by a significant margin.
>
> **"L170: "Intuitively, the NCE loss [ 23 ] pushes $p(s^{\prime} \mid s, z)$ to be a delta-like density function. However, since we define behaviors with $n$ steps, we want a wider distribution" The second sentence does not follow from the first. Even if $p(s^{\prime} \mid s, z)$ is deterministic, it can still be applied for $N$ steps, to yield an $N$-step behavior. Can the paper clarify these statements?"** \
> We updated the paper to clarify this sentence. In short, NCE involves a temperature term that controls the weight given to negative samples. A low-temperature term pushes the training dynamics to assign a lot of densities to a given positive pair.
>
> Since, there is not a single positive pair, but $n$ positive pairs (a skill is defined by $n$ steps), we need a high-temperature term (e.g., 0.5) so that a single positive pair is not assigned too much density.
>
> **"Can you explain the following inconsistency? L260 claims that setting Z_min = Z_max is equivalent to CIC, yet this results in different empircal performances in Fig 4b. By L260, we should expect their results to be equivalent. Why are they not? Or perhaps it's not equivalent to CIC, because it involves flipping the sign of the intrinsic reward function?"** \
> Yes you are right, it's not equivalent to CIC because it involves flipping the sign of the intrinsic reward function.
>
> **"By the paper's definition of surprise as the entropy of the joint distribution of state transitions, the method is not optimizing only surprise (since it's just one of the two terms of the mutual information). The method name then becomes something of a misnomer..."** \
> We agree with the reviewer that the optimization involves both terms. However, we named our method using the task of the RL agent, which is to maximize intrinsic reward - in our case, only involving surprise.
>
> ## Summary:
> - We revised and updated the paper with reference to URLB for the definitions of "Competence"-, "Data"-, and "Knowledge"-based methods.
> - We revised and updated the paper to fix the characterization of the objective function.
> - We revised and updated the paper to fix the ambiguity between the objective function and the optimization procedure.
> - We revised and updated the paper to include the zero-shot results of MOSS along with the fine-tuning curves in Appendix C3.
> - We revised and updated the paper to include ablations on the skill vector in Appendix C2.
> - We revised and updated the paper to include results from Adversarial Surprise to the main results.
> - We added MOSS results into Fig 4b.
> - We revised and updated the paper to clarify the statement in L170

---

> > ### Comment · Reviewer_w2pW · 2022-08-08
> > **Thank you for your responses**
> >
> > Thank you for your responses. They comprehensively addressed my concerns.

---

> ### Author Response · Authors · 2022-08-02
> **Response to Reviewer w2pW (part 1/2)**
>
> We want to thank the reviewer for the very detailed suggestions.
>
>
> ## Response:
>
> **"There is some undefined terminology. "Competence"-, "Data"-, and "Knowledge"-based methods."** \
> Thanks for the suggestions. We added a footnote to refer readers to the URLB paper for their definitions.
>
> **"Can you fix the characterization of the objective function?"** \
> We used the term "state-transition" to refer to $p(s^{\prime},s)$. We agree with the reviewer that this might be misleading as the term "state-transition" usually refers to $p(s^{\prime}\mid s)$. We updated the paper accordingly where we now refer to $p(s^{\prime},s)$ as the joint distribution of state-transitions and $p(s^{\prime}\mid s)$ as  state-transition.
>
> **"Can you fix the ambiguity in the connection between the objective function and the optimization procedure?"** \
> Yes, Alg 1 is correct. Our agent uses Eq. 9 (i.e., the KNN estimator) as the only intrinsic reward during pretraining. We clarified the ambiguity in footnote of page 6. In short, the implementation provided by CIC (https://github.com/rll-research/cic) is based on the arxiv version (https://arxiv.org/abs/2202.00161v2) which is an updated version of the ICLR version (https://openreview.net/forum?id=kOtkgUGAVTX). In particular, in the arxiv version, the intrinsic reward only relies on the KNN joint entropy estimates and NCE is only used to train $\phi_s$, $\phi_z$, and $\psi$.  While the ICLR version (page 5) includes the NCE term in the intrinsic reward.
>
> **"The experiments do not adequately illustrate the effect that finetuning has on the pretrained policy."** \
> Thanks for the suggestion. We added zero-shot results and finetuning learning curves in the appendix. In particular, Quadruped is an example of domain where the policy does not benefit significantly from finetuning (compared to Walker and Jaco). We note that this trend is not specific to MOSS, i.e., we observe a similar trend for CIC and Adversarial Surprise [18]. We present the zero-shot results of MOSS_min and MOSS_max on Quadruped in the table below
>
> |                 | Quadruped Jump | Quadruped run | Quadruped Stand | Quadruped Walk |
> | --------------- | -------------- | ------------- | --------------- | -------------- |
> | MOSS_min,frozen | 85.3±9.5       | 45±4          | 110±13          | 43±4           |
> | MOSS_max,frozen | 627±28         | 370±16        | 767±35          | 306±12         |
> | MOSS Finetuned  | 674±11         | 485±6         | 911±11          | 635±36         |
>
>
> **"The current ablation does not illustrate the importance of the choice of the structure of the $Z$ priors, which includes both the dimensionality $d$, as well as the event space."** \
> We agree with the reviewer that the configuration of the skill vector may depend on the environment and downstream task.
> We added the ablation suggested by the reviewer in Appendix C2. Below, we ran experiments for two domains: Quadruped and Walker and provide a table of the results.
> ||Continuous 1|Continuous 16|Continuous 64 (MOSS)|Discrete 1|
> |:-|:-|:-|:-|:-|
> |Walker Flip|827±50|934±12|729±40|672±9|
> |Walker Run|329±41|245±44|531±20|335±20|
> |Walker Stand|930±7|830±25|962±3|931±8|
> |Walker Walk|954±4|960±2|942±5|751±70|
> |Quadruped Jump|193±27|649±25|674±11|194±50|
> |Quadruped Run|167±17|444±26|485±6|162±20|
> |Quadruped Stand|306±32|834±33|911±11|267±38|
> |Quadruped Walk|158±19|569±49|635±36|217±23|
>
> We find that a continuous skill performs better than a discrete skill but not by a large margin. However, it seems that in general, a higher dimensional skill performs better, especially in Quadruped which has a higher dimensional state and action space than the Walker domain. Our results confirm the reviewer's intuition and suggest that the optimal skill vector is task-dependent.
> We note that hyper-parameters in MOSS (e.g., skill dimension) are not tuned and simply taken from CIC for fair comparison.

---

### Official Review · Reviewer_GiY2 · 2022-07-10

**Rating:** 5
**Confidence:** 3
**Soundness:** 3 good
**Presentation:** 3 good
**Contribution:** 2 fair

**Summary:**

The paper presents a method for unsupervised skill discovery via reinforcement learning. Unlike most prior methods which seek to either maximize or minimize surprise, this paper proposes to learn a mixture policy where one component of the mixture is trained to minimize surprise, and another component is trained to maximize surprise. The paper builds very closely on the work of Laskin et al [1] (CIC), and the proposed change to CIC is very small (just a one-line change in Algorithm 1). Essentially, the proposed method partitions the latent skill space into two parts, where one part is trained to maximize surprise, and the other part if trained to minimize surprise. Whether the agent chooses to sample from the surprise-maximizing part of the skill space or the surprise-minimizing part is somewhat arbitrary: the agent uses surprise maximizing skills for the first half of an episode, and surprise-minimizing skills for the second half of the episode.

The authors evaluate their method on the URLB benchmark and VizDoom tasks, and show that their method slightly outperforms CIC on both unsupervised learning metrics as well as downstream task performance. The main advantage of the method is that, unlike CIC, we do not need to pre-specify whether the agent should minimize or maximize surprise, and the algorithm is able to do well in environments that require either.

[1] Michael Laskin, Hao Liu, Xue Bin Peng, Denis Yarats, Aravind Rajeswaran, and Pieter Abbeel. CIC: Contrastive intrinsic control for unsupervised skill discovery, 2022.

**Questions:**

In the Section 5, you say “but performed well on the standing task, which is inherently a low entropy task.” → did you mean to say high-entropy task instead?


**Limitations:**

I don't think the limitations are particularly well-addressed in the paper. The main limitation mentioned was that the proposed method is only applicable to episode RL settings (and not reset-free settings), but this seems slightly orthogonal to the main issue of the paper. I think a more detailed discussion on how to make the objective switching more adaptive would make the limitations section much stronger.

**Strengths And Weaknesses:**

Strengths
- Simple, easy-to-implement method.
- A somewhat more general technique for unsupervised reinforcement learning as compared to prior methods.
- Writing quality is good.

Weaknesses
- The choice of when to switch between a surprise minimizing and maximizing objectives is somewhat ad-hoc. Authors investigate a slightly more sophisticated procedure in the Appendix, but find it to be very sensitive to hyperparameter choices, and omit it from the main paper.
- Overall writing quality is good, but the authors spend a lot of time (2.5 pages) going over background material and summarizing past work, while the main method section is about 1 page long, including an algorithm box. The main method section does not start until the second half of page 5! Given that the paper builds very closely on CIC, a more detailed exposition on CIC and limited discussion of other techniques would have sufficed.
- Some of the related work that also proposes to maximize and minimize surprise was only mentioned in the last section. I think this should be moved up to be in Section 3 instead, where other related works are discussed.

---

> ### Author Response · Authors · 2022-08-02
> **Response to Reviewer GiY2**
>
> We want to thank the reviewer for the detailed and insightful review.
>
>
> ## Response:
>
> **"Some of the related work that also proposes to maximize and minimize surprise was only mentioned in the last section. I think this should be moved up to be in Section 3 instead, where other related works are discussed."** \
> We followed the reviewer's suggestion and updated the paper accordingly. Because Section 3's title focuses on competence-based methods and the related work in the last section is not on competence-based methods, we moved this part into Section 4 to better clarify our novelty compared with other approaches.
>
> **"In the Section 5, you say “but performed well on the standing task, which is inherently a low entropy task.” → did you mean to say high-entropy task instead?"** \
> At first, we did mean that the standing task is inherently a low entropy task. For this task, after the agent gets upright in a short amount of time, the agent does not move much for the remaining of the episode, which means overall, the joint $p(s,s^{\prime})$ has low entropy.
>
> In this revision, we agree that this sentence was a bit confusing and not precise. Therefore, we removed this remark from the final version of the paper.
>
> **"Given that the paper builds very closely on CIC, a more detailed exposition on CIC and limited discussion of other techniques would have sufficed."** \
> We agree with the reviewer. In fact, this revision describes the objective function of CIC in Section 3.2. And, for clarity, Section 4 only contains content related to our contribution, while Sections 3 & 2 introduce the necessary background upon which MOSS builds. Furthermore, we believe that the brevity of Sec 4 is representative of the simplicity of our method, which we consider a strength.
>
> **"The choice of when to switch between a surprise minimizing and maximizing objectives is somewhat ad-hoc." AND "I don't think the limitations are particularly well-addressed in the paper. ... I think a more detailed discussion on how to make the objective switching more adaptive would make the limitations section much stronger."** \
> We agree with the reviewer's opinion on this issue and have updated the limitation section accordingly. Compared to more sophisticated methods (e.g., Appendix B) for setting $M$ (i.e., switching the objective), our heuristic does not increase the computational cost during training, does not require hyper-parameter tuning, and performs well.
>
> In our revised and updated paper, we proposed a meta controller [28] as possible future work. In particular, the meta controller would be trained to identify the optimal lower-level skill to perform, the steps each skill should take, and the objective to focus on. This meta controller will act according to higher-level abstractions and possibly incentivize structured deep exploration [43].
>
> ## Summary:
> - We moved the related work that was in the discussion section to Section 4.
> - We removed a confusing sentence from Section 5.
> - We updated the discussion & limitation section to deepen the discussion about objective switching.

---

> > ### Comment · Reviewer_GiY2 · 2022-08-04
> > **Thank you for the response!**
> >
> > Thank you for the response, I think these changes really improve the paper. However, since most of my comments/concerns that can addressed in the rebuttal period were relatively minor, and my major concern around how to switch objectives is something that can only addressed in future work, I plan to keep my original rating.

---

> > > ### Author Response · Authors · 2022-08-08
> > > **Follow-up Response to Reviewer GiY2**
> > >
> > > We appreciate the follow-up response from the reviewer. Although our objective switching mechanism appears to be ad-hoc, we consider its simplicity a strength for the following reasons: it does not increase the computational cost during training, does not require hyper-parameter tuning, and performs well. We agree that more sophisticated methods might perform better. However, we are not sure that the performance gain outweighs the possible increase in complexity (e.g., computational cost or hyper-parameter tuning.) We welcome any additional suggestions from the reviewer. Thanks!

---

### Official Review · Reviewer_zGJ4 · 2022-07-11

**Rating:** 6
**Confidence:** 4
**Soundness:** 3 good
**Presentation:** 3 good
**Contribution:** 3 good

**Summary:**

The paper proposes a new method called Mixture of Surprises (MOSS), which tries to maximize and minimize the surprise at the same time. This results in state-of-the-art performance on the URLB benchmark.

**Questions:**

Can this method also be used with demonstrations?

**Limitations:**

The authors should try to motivate the novelty of the paper a bit more.

**Strengths And Weaknesses:**

Strengths:
The paper is well written and covers the related literature quite well. The experiments are well done on variety of different environments. The proposed method also performs well on the URLB benchmark.


Weakness:
Already exist methods which try to maximize and minimize surprise at the same time. Thus, it is difficult to find novelty in the proposed method.

---

> ### Author Response · Authors · 2022-08-02
> **Response to Reviewer zGJ4**
>
> We want to thank the reviewer for the insightful feedback.
>
> ## Response
> **"Already exist methods which try to maximize and minimize surprise at the same time. Thus, it is difficult to find novelty in the proposed method." AND "The authors should try to motivate the novelty of the paper a bit more."** \
> Thanks for the suggestion. We have updated Section 4 accordingly to better clarify our novelty. The challenge is how to optimize these two opposite objectives. Thus, we consider that our novelty lies in the simplicity of how we maximize and minimize surprise simultaneously. In particular,
> - Prior work relies on an adversarial game [18] known to be challenging to train [37]. Instead, we propose a simple mixture of policies. Our implementation only requires a minor change to CIC.
> - Prior work builds upon an on-policy algorithm that requires training two policies (Alice and Bob) [18] that do not share parameters. Instead, our method builds upon an off-policy algorithm that uses a single network for both objectives, resulting in higher sample efficiency [b].
>
> In the table below, we report the experimental results of AS [18] to compare performance with MOSS on URLB and update the paper's main results (Figure 2 and Table 1).
>
> ||Walker Flip|Walker Run|Walker Stand|Walker Walk|Quadruped Jump|Quadruped Run|Quadruped Stand|Quadruped Walk|Jaco Bottom Left|Jaco Bottom Right|Jaco Top Left|Jaco Top Right|
> |:-|:-|:-|:-|:-|:-|:-|:-|:-|:-|:-|:-|:-|
> |MOSS|729±40|531±20|962±3|942±5|674±11|485±6|911±11|635±36|151±5|150±5|150±5|150±6|
> |AS-ALICE-4 MIL|491±20|211±9|868±47|655±36|415±20|296±18|590±41|337±17|109±20|141±19|140±17|129±15|
> |AS-ALICE-2MIL |469±13|195±14|914±17|655±33|380±18|279±9|543±21|276±19|114±20|158±15|143±13|120±20|
> |AS-BOB-4MIL|475±16|247±23|917±36|675±21|449±24|285±23|594±37|353±39|116±21|166±12|143±12|139±13|
> |AS-BOB-2MIL|474±23|223±15|937±15|687±42|427±20|262±8|590±22|314±23|118±21|168±15|154±13|121±14|
>
> We include the results for Alice and Bob since the original paper did not show how to choose the policy to finetune for downstream tasks. Additionally, since Alice and Bob are parameter-independent policies, we include the results for both pretrained using 2 mil and 4mil environment steps for easy comparison. From these experimental results, MOSS outperforms AS in URLB.
>
> **Can this method also be used with demonstrations?** \
> Thanks for the suggestion. Yes, we believe our method can be used with demonstrations. Unsupervised reinforcement learning methods (e.g., MOSS) are generally used to pretrain a policy (and critic) for faster adaption to downstream tasks [31]. If we already have demonstrations, the downstream task is probably known or can be inferred using inverse reinforcement learning methods.
> Unfortunately, we did not find a method that uses unsupervised reinforcement learning pretraining and then finetuned to demonstration data. Therefore, we did a proof-of-concept experiment by running the following demonstration experiment in the Walker domain in DMC: \
> We trained 3 differently seeded expert policies on each of the 4 tasks in the Walker domain. At the end of training, we constructed a dataset of 1 million state-action tuples by rolling them out for each seed. Finally, we trained using these datasets by behavior cloning using policy networks that are either (1) **Scratch**: randomly initialized or (2) **Pretrained**: pretrained using MOSS. We report each task's mean and standard error below (each seed is evaluated for 50 episodes).
>
> ||Walker Walk|Walker Run|Walker Flip|Walker Stand|
> |-|:-:|:-:|:-:|:-:|
> |**Scratch**|283.8±5.0|120.2±6.2|256.5±7.0|437.0±14.1|
> |**Pretrained**|312.2±4.6|119.5±0.3|261.6±3.1|463.1±9.9|
>
> We observe that pretrained policies perform better. Overall, we believe that a detailed investigation on this topic could be done for future work.
>
> Furthermore, recent work [a] used unsupervised reinforcement learning (URL) methods to collect data for offline reinforcement learning - a setting similar to demonstrations but with the addition of reward signals. The paper demonstrates that the dataset collected using URL methods is better than a dataset collected only using data from an expert trained using supervised RL. The paper argues that URL methods collect data samples that "maximize various notions of coverage or diversity" [a], which helps offline RL.
>
> ## Summary
> - We updated Sec 4 (L174-191) to clarify the novelty of our method in comparison to previous work.
> - We added results for Adversarial Surprise to the main results (Fig 2 and Table 1).
>
> ## References
> [a] Yarats, D., Brandfonbrener, D., Liu, H., Laskin, M., Abbeel, P., Lazaric, A., & Pinto, L. (2022). Don't Change the Algorithm, Change the Data: Exploratory Data for Offline Reinforcement Learning. arXiv preprint arXiv:2201.13425. \
> [b] Yarats, D., Zhang, A., Kostrikov, I., Amos, B., Pineau, J., & Fergus, R. (2021, May). Improving sample efficiency in model-free reinforcement learning from images. AAAI

---

### Official Review · Reviewer_x6bw · 2022-07-13

**Rating:** 5
**Confidence:** 4
**Soundness:** 2 fair
**Presentation:** 2 fair
**Contribution:** 2 fair

**Summary:**

The paper presents an improved algorithm based on CIC (Laskin et al. 2022), to balance surprise maximization and minimization for RL pertaining. Experiments on a variety of tasks show the effectiveness of the proposed methods.

**Questions:**

- It would be better to add MOSS results in Figure 4b) for direct comparison. It makes sense that MOSS_SAME performance is in between CIC and negative CIC, the question is why MOSS can do better. One would argue that in expectation MOSS is still just a linear interpolation of CIC and negative CIC (or a scaled CIC). Why is this not the case?
- How do you sample M? Suppose it's sampled from Bernoulli(p). Does setting p=0 give you CIC and settting p=1 give you negative CIC? What are the optimal p's for different environments? Would that somewhat correspond to environmental stochasticity based on your assumption?

**Limitations:**

Limitations and the potential negative impact were discussed at the end of the paper.

**Strengths And Weaknesses:**

The motivation of balancing maximizing and minimizing surprises is well explained, but it's unclear how the proposed method resolves the issue and how much it is different from a scaled CIC. I suggest the author make a clear distinction between MOSS objective and a linear interpolation of CIC and negative CIC (or equivalently a scaled CIC), and also show in the experimental section how the prior distribution of M affects MOSS performance.

---

> ### Author Response · Authors · 2022-08-02
> **Response to Reviewer x6bw**
>
> We want to thank the reviewer for the thoughtful feedback.
>
> ## Response:
> **"It would be better to add MOSS results in Figure 4b) for direct comparison."** \
> We added MOSS results in Figure 4b as suggested. Please see the updated paper.
>
> **"It's unclear how the proposed method resolves the issue"** \
> We resolved the issue of optimizing two opposite objectives using a mixture of policies, i.e., we view MOSS as a mixture of experts [a]. Since the mixture of policies builds on a competence-based approach, MOSS has two disjoint skill sets. MOSS separates the two opposite objectives using a different skill set for each objective: one skill set for surprise maximization and another for surprise minimization.
>
> **"I suggest the author make a clear distinction between MOSS objective and a linear interpolation of CIC and negative CIC (or equivalently a scaled CIC). It makes sense that MOSS_SAME performance is in between CIC and negative CIC; the question is why MOSS can do better. One would argue that in expectation, MOSS is still just a linear interpolation of CIC and negative CIC (or a scaled CIC). Why is this not the case?"** \
> We agree with the reviewer's understanding and modified the beginning of Section 4 accordingly to reflect the distinction between the MOSS objective and a linear interpolation of CIC and negative CIC (i.e., scaled CIC).
>
> We agree with the reviewer that MOSS_SAME is a linear interpolation of CIC and negative CIC. Indeed, in MOSS_SAME, we directly add the CIC and negative CIC objective, giving us an objective that still corresponds to either maximizing or minimizing.
>
> In contrast, MOSS' objective corresponds to maximizing and minimizing surprise simultaneously. Using a mixture of policies, we get two disjoint skill sets. We separate the two opposite objectives using a different skill set for each objective: one skill set for surprise maximization and another for surprise minimization. Indeed, our main contribution is this mixture of experts, and removing it from MOSS yields MOSS_SAME.
>
> **"How do you sample $M$?"** \
> In MOSS, we deterministically set $M=0$ first half of an episode and $M=1$ for the second half of the episode. Such a deterministic rule makes no assumptions about the environment's entropy. Compared to more sophisticated methods (e.g., Appendix B) for setting $M$, our heuristic does not increase the computational cost during training, does not require hyper-parameter tuning, and performs well.
>
> **"Suppose $M$ is sampled from $\operatorname{Bernoulli}(p)$. Does setting $p=0$ give you CIC and setting $p=1$ give you negative CIC? What are the optimal p's for different environments? Would that somewhat correspond to environmental stochasticity based on your assumption?"** \
> We agree with the reviewer that If $M$ is a Bernoulli, then setting $p=0$ gives CIC, and $p=1$ gives Negative CIC.
>
> We also agree with the reviewer's comments on $p$. The optimal $p$ tends to align with our assumption of the environment. For example, in the URLB environment, we tried sampling with $p=0.3$ (biased towards entropy minimization), and the performance dropped drastically compared to $p=0.6$. However, we emphasize that in our final method, we did not sample $M$ from a Bernoulli because we observed a higher variance across seeds compared to the deterministic rule.
>
>
> **"Show in the experimental section how the prior distribution of $M$ affects MOSS performance"** \
> Thanks for the suggestion. Due to the lack of space, we added the suggested experiments in Appendix C1.  In MOSS, we deterministically set $M=0$ for the first half of the steps and $M=1$ for the other half of the steps in the episode. This corresponds to having $50$% of maximization data and $50$% of minimization data. Below, we report results for different ratios of maximization and minimization where 50% corresponds to MOSS:
>
> |                  | Walker Flip | Run    | Stand | Walk   |
> | ---------------- | ----------- | ------ | ----- | ------ |
> | 30% maximization | 723±36      | 515±32 | 967±3 | 921±18 |
> | 40%              | 738 ±42     | 526±15 | 968±2 | 908±24 |
> | 50% (MOSS)       | 729±40      | 531±20 | 962±3 | 942±5  |
> | 60%              | 781±38      | 599±11 | 967±1 | 935±6  |
> | 70%              | 785±39      | 567±15 | 965±3 | 933±7  |
>
> In short, on Walker, a prior towards entropy maximization results in a better performance than a prior towards entropy minimization.
>
>
> ## Summary
> - We added MOSS results into Fig 4b.
> - We modified Section 4 (L174) to clarify:\
>  (1) how the proposed method resolves the issue of optimizing two opposite objectives. \
>   (2) the distinction between MOSS objective and a scaled CIC.
> - We added ablations on the prior distribution of $M$ in Appendix C1.
>
> ## References
> [a] https://en.wikipedia.org/wiki/Mixture_of_experts

---

### Author Response · Authors · 2022-08-02
**Summary of Changes**

Thank you to everyone involved in this reviewing process. We appreciate all four reviewers' insightful comments on our submission. We have carefully addressed all reviewers' comments and responded back to each reviewer separately. We also uploaded a revised submission that incorporated all reviewers' suggestions. To ease the reviewing process, we summarize the major changes below. We welcome further comments and suggestions that improve the quality of our submission. Thank you!
- We added MOSS results in Fig 4b.
- We modified Section 4 (L174) to clarify :
  - the novelty of our method in comparison to previous work.
  - the distinction between MOSS objective and a scaled CIC.
- We added ablations on the prior distribution of $M$ in Appendix C1.
- We added results for Adversarial Surprise to the main results (Fig 2 and Table 1).
- We moved the related work that was in the discussion section to Section 4.
- We moved the benchmark render figures to the appendix (Figure 4).
- In Section 5, we removed a confusing sentence.
- We revised and updated the discussion & limitation section to deepen the discussion about objective switching.
- We revised and updated the paper with reference to URLB to clarify the definitions of "Competence"-, "Data"-, and "Knowledge"-based methods.
- We revised and updated the paper to fix the characterization of the objective function.
- We revised and updated the paper to fix the ambiguity between the objective function and the optimization procedure.
- We revised and updated the paper to include the zero-shot results of MOSS along with the fine-tuning curves in Appendix C.
- We revised and updated the paper to include ablations on the skill vector in Appendix C2.
- We revised and updated the paper to clarify the statement in L170 of the previous revision.

Finally, for citations that appear in the rebuttal, numerical citations match the reference numbers in the pdf with the appendix; alphabetical citations are newly added references for each reply and are listed at the end of the reply.

---

### Meta-Review · Area_Chair_woEu · 2022-08-24

**Recommendation:** Accept
**Confidence:** Certain

**Metareview:**

This paper proposes a method for unsupervised skill discovery, which learns a mixture of policies that simultaneously maximizes and minimizes the surprise. All the reviewers agree that the paper tackles an important active research area. The paper is well written; the motivation is well explained; the proposed method is simple and easy-to-implement; and it performs well on the benchmark. Several concerns were raised by the reviewers, including novelty, delta beyond the interpolated CIC, and the choice of M. The rebuttal and the additional experimental results have addressed some of these concerns. One of the main criticisms, choice on when to switch the objectives, still remains after the rebuttal. Although the current heuristic is simple, it seems ad-hoc, without good justifications. After  discussion with reviewers, we agree that this limitation is compensated by the other contributions of the paper, and helps to set the stage for future work that does optimize M or devises a new method that does not rely on it. Thus, we recommend accepting this paper.

**Award:**

No

---

### Decision · Program_Chairs · 2022-09-14

Accept